# Predicting Isocitrate Dehydrogenase Status in Non-Contrast-Enhanced Adult-Type Astrocytic Tumors Using Diffusion Tensor Imaging and ^11^C-Methionine, ^11^C-Choline, and ^18^F-Fluorodeoxyglucose PET

**DOI:** 10.3390/cancers16081543

**Published:** 2024-04-18

**Authors:** Shoji Yasuda, Hirohito Yano, Yuka Ikegame, Soko Ikuta, Takashi Maruyama, Morio Kumagai, Yoshihiro Muragaki, Toru Iwama, Jun Shinoda, Tsuyoshi Izumo

**Affiliations:** 1Department of Neurosurgery, Chubu Medical Center for Prolonged Traumatic Brain Dysfunction, Minokamo 505-0034, Japan; hirohito@herb.ocn.ne.jp (H.Y.); ikegame-nsu@umin.ac.jp (Y.I.); mor123155@yahoo.co.jp (M.K.); junshino@joy.ocn.ne.jp (J.S.); 2Department of Neurosurgery, Chubu Neurorehabilitation Hospital, Minokamo 505-0034, Japan; 3Department of Neurosurgery, Gifu University Graduate School of Medicine, Gifu 501-1194, Japan; go-izumo@hotmail.co.jp; 4Department of Clinical Brain Sciences, Gifu University Graduate School of Medicine, Gifu 501-1194, Japan; 5Department of Neurosurgery, Tokyo Women’s Medical University, Tokyo 162-8666, Japan; sikuta@twmu.ac.jp (S.I.); mtm2727@gmail.com (T.M.); ymuragaki@twmu.ac.jp (Y.M.); 6Department of Neurosurgery, Gifu Municipal Hospital, Gifu 500-8513, Japan; toruiwama1776@gmail.com

**Keywords:** glioma, positron emission tomography, diffusion tensor imaging, isocitrate dehydrogenase mutation, astrocytoma, glioblastoma

## Abstract

**Simple Summary:**

Predicting the isocitrate dehydrogenase (IDH) mutation status is difficult in preoperative non-enhanced astrocytic tumors. We aimed to differentiate IDH status using preoperative MRI and PET. We found that patients with IDH-mutant (mut) astrocytomas were significantly younger than those with IDH-wildtype (wt) glioblastomas. The tumor location, fractional anisotropy, and mean diffusivity on MRI were significantly related to the IDH mutation status. PET showed significantly higher uptakes for ^11^C-methionine, ^11^C-choline, and ^18^F-fluorodeoxyglucose in IDH-wt than in IDH-mut tumors. Together, these findings reflect the higher malignancy of IDH-wt than of IDH-mut tumors. Composite diagnosis using age, MRI, and PET findings resulted in high accuracy for predicting IDH status. Thus, approaches based on biological tumor behavior in early-stage gliomas are helpful for guiding preoperative treatment decisions.

**Abstract:**

We aimed to differentiate the isocitrate dehydrogenase (IDH) status among non-enhanced astrocytic tumors using preoperative MRI and PET. We analyzed 82 patients with non-contrast-enhanced, diffuse, supratentorial astrocytic tumors (IDH mutant [IDH-mut], 55 patients; IDH-wildtype [IDH-wt], 27 patients) who underwent MRI and PET between May 2012 and December 2022. We calculated the fractional anisotropy (FA) and mean diffusivity (MD) values using diffusion tensor imaging. We evaluated the tumor/normal brain uptake (T/N) ratios using ^11^C-methionine, ^11^C-choline, and ^18^F-fluorodeoxyglucose PET; extracted the parameters with significant differences in distinguishing the IDH status; and verified their diagnostic accuracy. Patients with astrocytomas were significantly younger than those with glioblastomas. The following MRI findings were significant predictors of IDH-wt instead of IDH-mut: thalamus invasion, contralateral cerebral hemisphere invasion, location adjacent to the ventricular walls, higher FA value, and lower MD value. The T/N ratio for all tracers was significantly higher for IDH-wt than for IDH-mut. In a composite diagnosis based on nine parameters, including age, 84.4% of cases with 0–4 points were of IDH-mut; conversely, 100% of cases with 6–9 points were of IDH-wt. Composite diagnosis using all parameters, including MRI and PET findings with significant differences, may help guide treatment decisions for early-stage gliomas.

## 1. Introduction

The 2021 revision of the World Health Organization (WHO) classification of central nervous system tumors reclassified adult diffuse gliomas into the following three types: oligodendrogliomas, isocitrate dehydrogenase mutant (IDH-mut), and 1p/19q co-deleted; astrocytomas, IDH-mut; and glioblastomas, IDH-wildtype (wt). The clinical outcomes and prognoses of these three glioma types differ greatly depending on the genetic diagnosis, and various methods for predicting the tumors’ pathological characteristics using preoperative imaging features have been reported. Astrocytomas and glioblastomas, classified as astrocytic tumors without a 1p/19q co-deletion, are distinguished using the IDH status; however, they are not always accurately diagnosed using preoperative findings. 

The median overall survival of patients with glioblastomas of WHO classification Grade 4 is eight months, and the 5-year survival rate is 6.9% (or even 9.8% after optimal treatment) [1,2]. Conversely, early-stage astrocytomas of WHO classification Grade 2, which are referred to as diffuse astrocytomas, IDH-mut in the 2016 WHO classification, lack the histological features of anaplasia with less or no mitotic activity and have a promising long-term prognosis.

The use of MRI, contrast patterns, tumor locations, fractional anisotropy (FA), and mean diffusivity (MD) values from diffusion tensor imaging (DTI) has been reported for the differential diagnosis between astrocytomas and glioblastomas [3,4,5,6]. Our institution has also reported on the usefulness of PET using ^11^C-methionine (MET), ^11^C-choline (CHO), and ^18^F-fluorodeoxyglucose (FDG) for differentiating among gliomas (particularly those with IDH mutations) [7,8]. However, only a few reports have addressed non-contrast-enhanced lesions that require more careful clinical differentiation [9], and the differentiation of such lesions is often difficult. Astrocytomas (IDH-mut) and glioblastomas (IDH-wt) have different features in growth rate and prognosis [10,11,12]. Therefore, their preoperative diagnosis is essential for determining whether to perform surgery (such as tumor resection or biopsy) as well as maintaining the balance between the tumor removal rate and neurological sequelae (considering postoperative therapy, such as chemotherapy or radiotherapy) in cases where surgery is required.

In this study, we attempted to diagnose the IDH status of non-contrast-enhanced astrocytic tumors using MRI and PET.

## 2. Materials and Methods

### 2.1. Patients

We retrospectively analyzed the data of patients with non-contrast-enhanced, diffuse, supratentorial astrocytic tumors who underwent MRI and PET at our hospital between May 2012 and December 2022. 

During the study period, 527 patients with preoperative gliomas underwent MRI and PET examinations; 241 patients received their pathological diagnoses. We excluded the following patients (*n* = 159): those with contrast-enhanced lesions, oligodendrogliomas, or brain stem or cerebellar gliomas; those aged < 18 years; and those whose IDH status was not evaluated. Therefore, we finally included 82 patients in this study.

We classified the IDH status via immunohistochemistry using anti-IDH1 R132H according to previously reported methods [8] and confirmed the absence of the 1p/19q co-deletion via fluorescence in situ hybridization to exclude oligodendrogliomas. According to the 2021 WHO Classification, 55 and 27 patients had non-contrast-enhanced astrocytomas (IDH-mut) and glioblastomas (IDH-wt), respectively. 

This study was approved by the Institutional Ethics Review Committee of the Chubu Neurorehabilitation Hospital (Minokamo City, Japan; approval no. 2023-05) and was performed following the principles of the 1964 Declaration of Helsinki and its later amendments and comparable ethical standards. The requirement of obtaining informed consent was waived because of the study’s retrospective design. Patients could opt out after referring to the information disclosure document available on our hospital’s website (https://cnrh.jp/news/upload/32-0link_file.pdf, (accessed on 16 March 2024)).

### 2.2. MRI Procedure

MRI was performed using an Achieva 3.0 T TX QD MRI system (Phillips, Amsterdam, The Netherlands). A single-shot spin-echo echo-planar sequence was used for DTI (diffusion encoding directions, 6; repetition time [T.R.], 7000 ms; echo time [T.E.], 70 ms; flip angle, 70°; field of view [FOV], 256 mm; matrix size, 128 × 128 mm; b-values, 0–1000 s/mm^2^; slice thickness, 2 mm; and total slices, 70). T1-weighted images were obtained with the following parameters: T.R., 2200 ms; T.E., 9.5 ms; flip angle, 90°; FOV, 230 × 230 mm; matrix size, 512 × 512 mm; and slice thickness, 5 mm (with a 1 mm slice gap). For contrast-enhanced sequences, patients were injected with a gadolinium-based contrast agent; gadoteridol was chosen for patients weighing less than 80 kg, whereas gadoterate meglumine was selected for patients weighing over 80 kg. 

The high-intensity regions in T2-weighted images (T2WI) were defined as lesions in which the tumor distribution had to be assessed. We evaluated whether the lesions were confined to a single lobe or had spread to multiple lobes; whether they invaded the surrounding structures, such as the insula, thalamus, corpus callosum, or contralateral cerebral hemisphere; and whether they were adjacent to the ventricle walls. Three or more neurosurgeons blinded to the study judged the tumor location. FA and MD values were calculated from DTI using the Dr. View/Linux software R2.5.0 (AJS Corporation, Tokyo, Japan). The ROI for FA and MD values was defined as a high-intensity area on an axial b0 image in three slices, including the slice with the largest diameter and the preceding and subsequent slices. The first author drew the ROI, and the mean FA and MD values were calculated.

### 2.3. PET Procedure

In our institute, PET evaluations using MET, CHO, and FDG are performed routinely for patients with brain tumors. PET was performed using Eminence STARGATE (Shimadzu Corporation, Kyoto, Japan), equipped with a 3D acquisition system. This system provides 99 transaxial images at 2.65 mm intervals, and the in-place spatial resolution (full width at half-maximum) is 4.8 mm. Axial images were adjusted parallel to the canthomeatal line. We injected 3.5 MBq/kg for all tracers and obtained the MET-, CHO-, and FDG-PET images in this order, with an interval of approximately 30–40 min between each shot to allow for radiation decay. The accumulation of each tracer was analyzed using the first author’s standardized uptake value (SUV). We calculated the tumor maximum SUV/normal frontal cortex SUV (T/N) ratio using the Dr. View/Linux software (AJS Corporation, Tokyo, Japan).

### 2.4. Data Analysis

We used Pearson’s chi-squared test and the Mann–Whitney *U* test to analyze categorical and continuous variables, respectively, to extract parameters showing significant differences concerning the IDH status. We calculated the cut-off value, sensitivity, specificity, and area under the curve (AUC) using receiver operating characteristic curves for continuous variables that showed significant differences. All statistical analyses were performed using the JMP statistical software 17.1.0 (SAS Institute Inc., Cary, NC, USA). Statistical significance was set at *p* < 0.05. Using the parameters we obtained, we calculated the ratios of the IDH status for each number of matching items to evaluate the diagnostic accuracy.

## 3. Results

A clinical summary of the 82 cases is presented in Table 1. The mean ages (±standard deviation) of patients with astrocytomas and those with glioblastomas were 38.0 ± 11.0 years and 52.1 ± 15.5 years, respectively (*p* < 0.0001); the proportions of male patients in these two groups were 58.2% (*n* = 32) and 37.0% (*n* = 10), respectively (*p* > 0.05; Table 1). The patients with astrocytomas were significantly younger than those with glioblastomas; the cut-off value, AUC, sensitivity, and specificity were 48, 0.76, 0.63, and 0.85, respectively. Grades 2, 3, and 4 were observed in 18.2% (*n* = 10), 80% (*n* = 44), and 1.8% (*n* = 1) of patients with astrocytomas, respectively. The intervals from PET and MRI examination to the operation were shorter for the patients with astrocytomas than in those with glioblastomas (*p* = 0.008). The lesions were confined to a single lobe in 67.3% (*n* = 37) and 59.3% (*n* = 16) of the patients with astrocytomas and those with glioblastomas, respectively (*p* > 0.05; Table 1). Invasion into multiple lobes was observed in 30.9% (*n* = 17) and 37.0% (*n* = 10) of the patients with astrocytomas and those with glioblastomas, respectively (*p* > 0.05; Table 1). The insula and corpus callosum involvement did not differ significantly between the two tumor types. However, tumors involving the thalamus (*p* = 0.006) and those reaching the contralateral cerebral hemisphere (*p* = 0.018) were significantly more frequent in patients with glioblastomas than in those with astrocytomas, respectively (Table 1). 

High-intensity areas adjacent to the ventricle walls on T2WI images, defined as lesions in this study, were significantly more frequent in patients with glioblastomas than in those with astrocytomas (*p* = 0.022; Table 1). The FA values were significantly higher in patients with glioblastomas than in those with astrocytomas (IDH-wt vs. IDH-mut groups: 0.20 ± 0.04 vs. 0.16 ± 0.03; *p* < 0.0001; Figure 1A); the cut-off value, AUC, sensitivity, and specificity were 0.18, 0.78, 67%, and 82%, respectively (Figure 1B). The MD values were significantly higher in patients with astrocytomas than in those with glioblastomas (IDH-mut vs. IDH-wt groups: 1.47 ± 0.21 × 10^−3^ vs. 1.24 ± 0.21 × 10^−3^; *p* < 0.0001; Figure 1C); the cut-off value, AUC, sensitivity, and specificity were 1.28, 0.78, 74%, and 80%, respectively (Figure 1D).

PET examinations revealed significant differences among all tracers: MET, CHO, and FDG accumulations were significantly higher in glioblastomas (IDH-wt) than in astrocytomas (IDH-mut). The MET T/N ratios were 1.51 ± 0.48 and 1.88 ± 0.80 in the IDH-mut and IDH-wt groups, respectively (*p* = 0.013; Figure 2A). The cut-off value, AUC, sensitivity, and specificity were 1.29, 0.65, 85%, and 44%, respectively (Figure 2B). The CHO T/N ratios were 1.44 ± 0.63 and 1.98 ± 0.86 in the IDH-mut and IDH-wt groups, respectively (*p* = 0.024; Figure 2C). The cut-off value, AUC, sensitivity, and specificity were 2.02, 0.66, 52%, and 95%, respectively (Figure 2D). The FDG T/N ratios were 0.79 ± 0.22 and 0.92 ± 0.24 in the IDH-mut and IDH-wt groups, respectively (*p* = 0.019; Figure 2E). The cut-off value, AUC, sensitivity, and specificity were 0.82, 0.65, 67%, and 64%, respectively (Figure 2F).

### 3.1. Diagnostic Accuracy Based on Each Parameter with Significant Differences (Figure 3)

Diagnostic accuracy was evaluated by combining parameters with significant differences. We scored five parameters (invasion into the thalamus, invasion into the contralateral cerebral hemisphere, location adjacent to the ventricle walls, FA value, and MD value) on MRI (Figure 3A,B). Astrocytomas (IDH-mut) were observed in 84.6% (*n* = 22/26) of the cases with 0 points, 82.1% (*n* = 46/56) of the cases with 0–1 points, and 82.8% (*n* = 53/64) of the cases with 0–2 points. Glioblastomas (IDH-wt) were observed in 88.9% (*n* = 16/18) of the cases with 3–5 points, 85.7% (*n* = 6/7) of the cases with 4–5 points, and 100% (*n* = 2/2) of the cases with 5 points. When the patients were divided into groups of 0–2 points and 3–5 points, the diagnostic accuracy was 84.1% (*n* = 69/82). Next, we evaluated the diagnostic accuracy of PET (Figure 3A,C). For scoring, we used three cut-off values of T/N ratios that were significantly higher in glioblastomas (IDH-wt). Astrocytomas (IDH-mut) were observed in 100% (*n* = 6/6) of the cases with 0 points and 83.0% (*n* = 44/53) of the cases with 0–1 points; conversely, glioblastomas (IDH-wt) were observed in 62.1% (*n* = 18/29) of the cases with 2–3 points and 80.0% (*n* = 8/10) of the cases with 3 points. When the patients were divided into groups of 0–1 points and 2–3 points, the diagnostic accuracy was 75.6% (*n* = 62/82).

We then scored all nine parameters, including age, with significant differences to evaluate the diagnostic accuracy (Figure 3A,D). Astrocytomas (IDH-mut) were diagnosed in 84.4% (*n* = 54/64) of the cases with 0–4 points, while glioblastomas (IDH-wt) were diagnosed in 100% (*n* = 15/15) of the cases with 6–9 points. The three cases with 5 points included one case of astrocytoma (IDH-mut) and two glioblastoma (IDH-wt) cases. When the patients were divided into groups of 0–4 and 6–9 points, the diagnostic accuracy was 87.3% (*n* = 69/79).

### 3.2. Representative Cases (Figure 4)

We present two representative cases based on composite diagnosis using the parameters with significant differences. Case 1 is a 34-year-old male with an astrocytoma, IDH-mut (Figure 4A), and case 2 is a 48-year-old female with a glioblastoma, IDH-wt (Figure 4B).

## 4. Discussion

In this study, we attempted to accurately diagnose the IDH status in astrocytic tumors by combining MRI and PET findings. Compared with contrast-enhanced gliomas, non-contrast-enhanced gliomas are less malignant and at an earlier tumor stage. A non-contrast-enhanced astrocytoma (IDH-mut) is considered a WHO classification Grade 2 equivalent (lacking necrosis and microvascular proliferation) and has a relatively favorable prognosis. In contrast, glioblastoma (IDH-wt) is known to show rapid progression and a poor prognosis. However, preoperative differential diagnosis between astrocytomas (IDH-mut) and glioblastomas (IDH-wt) in the early stages without enhancement effects is not always easy, and no established diagnostic imaging guidelines exist. The decision to administer surgical treatment to patients with asymptomatic non-contrast-enhanced gliomas may be delayed depending on age, comorbidities, or tumor localization. Therefore, the accurate preoperative diagnosis of IDH status is beneficial in clinical situations.

The differences in the clinical characteristics between astrocytomas and glioblastomas suggest that these tumors originate from different precursor cells [13]. Many studies have investigated the cell origin of glioblastomas; the possibility that subventricular zone cells may serve as the origin of glioblastomas has been reported recently [14,15,16]. The present study showed that high-signal areas, defined as lesions on T2WI, were adjacent to the ventricles in a significantly higher number of glioblastomas. Although we could not clearly determine whether this was due to periventricular edema or cellular infiltration, our results suggest that glioblastomas are associated with the ventricular wall.

DTI yielded higher FA and lower MD values for glioblastomas than for astrocytomas. This versatile, quantitative diagnostic method is widely used in clinical practice. In previous reports, the measurers often determined the ROI subjectively (e.g., only for the solid components) and not objectively for tumors without clear boundaries [17,18,19]. For the non-contrast-enhanced lesions in the present study, it was difficult to distinguish between the tumors and edematous changes clearly; objectivity was ensured by defining the entire high-intensity area in b0 images as the ROI. The trend wherein the FA and MD values were strongly correlated with the glioma molecular subtypes and acceptable AUCs, as in previous studies [6], was also observed in the present study, thereby confirming the method’s validity.

The FA value represents the strength of anisotropy based on tensor imaging in the range of 0–1, with higher values indicating more significant anisotropy. Higher FA values are correlated with a higher cell density [20] and are assumed to be related to a symmetrical histological organization (such as pseudopalisading necrosis, endothelial proliferation, or glomerular formation) [6,21]. However, the present study included only non-contrast-enhanced lesions in the early stage, generally not at the necrosis stage, with the upregulation of pro-angiogenic factors such as the vascular endothelial growth factor [22,23]. Glioblastomas (IDH-wt) may have demonstrated a high invasive potential to expand while sparing existing nerve fibers. 

In contrast, astrocytomas (IDH-mut) showed a significantly lesser invasion into the thalamus and contralateral cerebral hemisphere and less frequently reached the ventricular wall; this indicates their more localized nature. This trend suggests that compared with astrocytomas, glioblastomas have a higher invasive potential. MD values approximate the apparent diffusion coefficient values, and lower values indicate a higher cell density; thus, glioblastomas show a higher proliferative capacity owing to their higher cellularity than astrocytomas [4,6,24].

PET studies can reflect the biological metabolic activity; therefore, we used MET, CHO, and FDG to evaluate the tumor behavior. MET accumulation in gliomas reflects the proliferative activity, spread, and tumor grade, which can be explained through three mechanisms: active transport, passive diffusion, and stagnation [7,25]. CHO accumulation is controlled through amino acid transporter expression and attenuation in tumor endothelial cells. It coincides with lesion enhancement due to blood–brain barrier disruption, resulting in the leakage of the contrast agent [26,27]. FDG is less effective than MET for evaluating malignant gliomas because the high background uptake of FDG hinders tumor uptake evaluation. However, ^18^F has a longer half-life than ^11^C, providing greater versatility. Recently, in revealing the effectiveness compared with FDG, other amino acid tracers using ^18^F, such as fluoroethyltyrosine or fluorodihydroxyphenylalanine, are expected to define the tumor extent or malignancy of glioma in practice. L-amino acid transporters are broadly expressed in tumor tissue, which is not normal brain tissues [28]. The accumulation ratios of MET, CHO, and FDG in both groups were lower in this study, which included only non-contrast-enhanced gliomas, than those reported in previous studies, which included gliomas regardless of them being contrast-enhanced [8,29,30]. This suggests that compared with contrast-enhanced lesions, non-enhanced lesions are in the earlier stages of the disease and have a lower biological activity; however, glioblastomas showed a significantly higher accumulation of all three tracers than astrocytomas, indicating their high malignant potential from the earliest stages of the disease. Moreover, the significantly higher uptake of CHO in glioblastomas suggests that the tumor may rapidly progress to contrast-enhanced lesions. Changes to the contrast enhancement effects may have been observed if a glioblastoma case had a wait-and-see procedure.

The eight identified parameters with significant differences, except for age, suggest that compared with astrocytomas, glioblastomas have a higher biological activity, characterized by their invasive and proliferative capacities; this indicates their malignant potential in the early stages, leading to a precise diagnosis of the IDH status. A clear distinction between astrocytomas and glioblastomas is clinically significant and can lead to the appropriate evaluation of treatment strategies.

In the ROC analysis for IDH status identification, age was a crucial factor because its AUC was higher than those of the three PET markers. We analyzed age as the ninth parameter that distinguishes IDH statuses; however, it did not significantly improve diagnostic accuracy, possibly due to the presence of several factors, and the impact of each was offset. 

We have previously reported on identifying gliomas using PET [7,8]; however, PET findings alone were insufficient to achieve an accurate diagnostic yield in this study. Although the eight parameters that showed significant differences showed insufficient AUCs alone, their combination yielded a higher positive diagnosis rate. Recent advances in genetic analysis have led to identifying genes in astrocytomas that may be associated with a poor prognosis [31]. Genetic aberrations associated with worse prognoses, such as *PDGFRA* amplification, *CDKN2A* or *CDKN2B* deletion, and *CDK4* amplification, may lead to further glioma subdivision. In this regard, this noninvasive method may be helpful in identifying high-risk groups of astrocytomas in the early stages.

It is essential to recognize the imaging pattern resulting from the differences in the biological behavior of each tumor. Although several studies have referred to IDH mutation using M.R. spectroscopy, perfusion, permeability, or chemical exchange saturation transfer imaging, using solely one factor may not suffice for non-contrast-enhanced gliomas [32,33,34,35,36,37]. As molecular diagnosis is becoming greatly important in glioma treatment, a versatile and simple method for predicting genetic differences remains warranted. Our results will help to facilitate the development of preoperative diagnostic methods at the earliest stage of astrocytic tumors.

This study has several limitations. First, the lesion defined to calculate the FA and MD was based on a b0 image similar to a T2WI. Although fusing the b0 image with a T2WI or FLAIR image might provide a more accurate analysis, this fusion is technically challenging when performing a DTI analysis, which makes it unfeasible for clinical practice. Second, the presence of IDH mutations was confirmed using immunohistochemistry, which is not exhaustive. For cases involving patients aged > 55 years, if immunohistochemistry ruled out IDH1 mutation, no other sequencing may be required [38,39]. However, we do not deny the possibility of IDH2 mutation in patients aged ˂ 55 years. In future studies, comprehensive DNA sequencing should be performed in all cases for a more accurate imaging diagnosis. Finally, the calculated diagnostic accuracy was based on the parameters analyzed from the same dataset. Although the significant parameters predicting IDH mutation confirmed the diagnostic accuracy in another validation dataset, such validation was impossible since glioma is not a common disease, particularly non-contrast-enhanced glioblastoma (IDH-wt). However, this study proposes several aspects to consider regarding tumor characteristics. Nonetheless, a prospective study remains warranted to confirm the validity of our results.

## 5. Conclusions

Non-contrast-enhanced astrocytic tumors can be differentiated based on the IDH status with high accuracy using a combination of MRI and PET. The presumption of a genetic diagnosis from the earliest stages of tumor development allows for earlier interventions in high-risk cases without contrast-enhancement effects. It is essential to avoid the misidentification of IDH-wt in the imaging diagnosis of gliomas at their early stages.

## Figures and Tables

**Figure 1 cancers-16-01543-f001:**
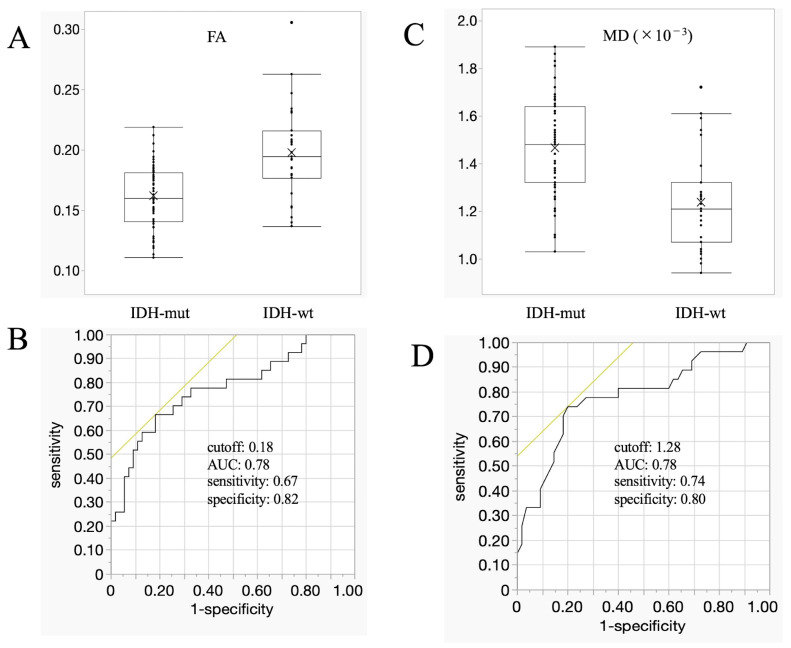
MRI findings. (**A**,**C**) Box plots showing the mean FA (**A**) and MD (**C**) values, compared between astrocytomas (IDH-mut) and glioblastomas (IDH-wt). Compared with astrocytomas, glioblastomas show significantly higher FA and significantly lower MD values (*p* < 0.0001 for both). Lines within the boxes indicate the median, the boxes represent the interquartile range, the whiskers denote the minimum and maximum, and the cross marks indicate the mean. (**B**,**D**) ROC curves showing the cut-off values, AUCs, sensitivities, and specificities of the FA (**B**) and MD (**D**) values. Yellow lines are tangent line at cutoff value. AUC, area under the curve; FA, fractional anisotropy; IDH, isocitrate dehydrogenase; MD, mean diffusivity; mut, mutant; ROC, receiver operating characteristic; wt, wildtype.

**Figure 2 cancers-16-01543-f002:**
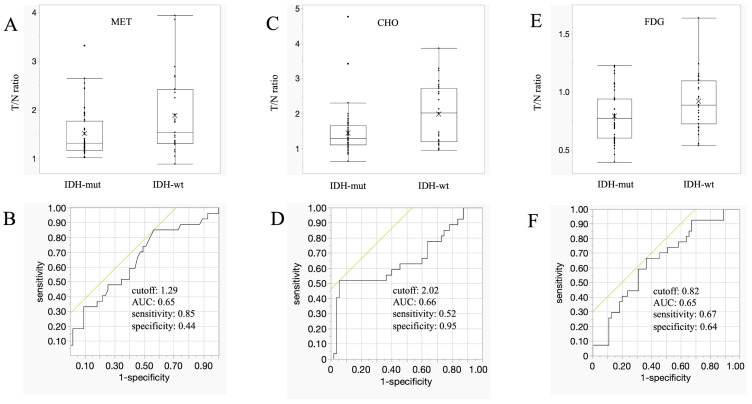
PET findings. (**A**,**C**,**E**): Box plots showing the mean T/N ratios, compared between astrocytomas (IDH-mut) and glioblastomas (IDH-wt) for MET (**A**), CHO (**C**), and FDG (**E**). Accumulations of all tracers were significantly higher in glioblastomas than in astrocytomas (MET, *p* = 0.0128; CHO, *p* = 0.024; FDG, *p* = 0.00185). Lines within the boxes indicate the median, boxes represent the interquartile range, whiskers denote the minimum and maximum, and cross marks indicate the mean. (**B**,**D**,**F**) ROC curves showing the cut-off values, AUCs, sensitivities, and specificities for MET (**B**), CHO (**D**), and FDG (**F**). Yellow lines are tangent line at cutoff value. AUC, area under the curve; CHO, ^11^C-choline; FDG, ^18^F-fluorodeoxyglucose; IDH, isocitrate dehydrogenase; MET, ^11^C-methionine; mut, mutant; ROC, receiver operating characteristic; T/N ratio, tumor/normal brain ratio; wt, wildtype.

**Figure 3 cancers-16-01543-f003:**
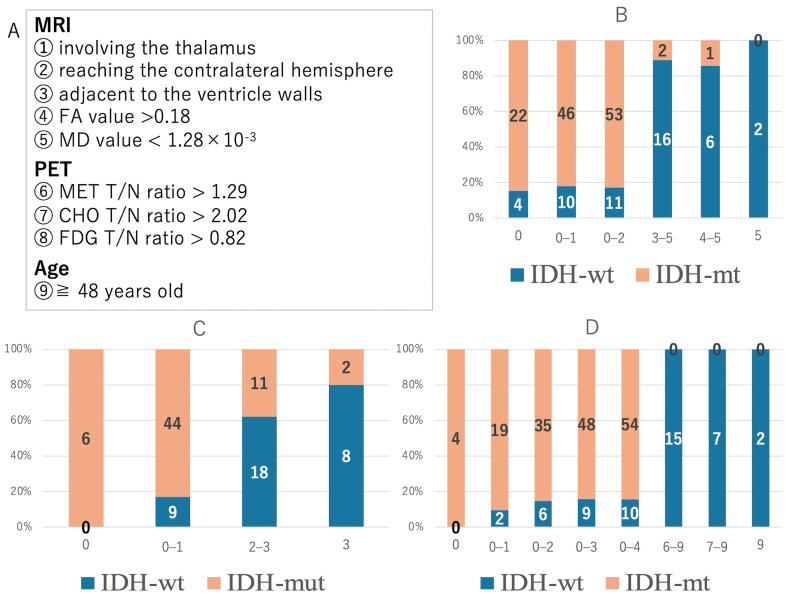
Diagnostic accuracy based on each parameter with significant differences. The numbers in the graph represent the patients matched by score. (**A**) The parameters indicating IDH-wildtype in MRI findings, PET findings, and age. (**B**,**C**) The graphs show the diagnostic accuracy based on the parameters detected on MRI (**B**) and PET (**C**). (**D**) The composite diagnostic methods resulting from nine parameters, including age, show high accuracy.

**Figure 4 cancers-16-01543-f004:**
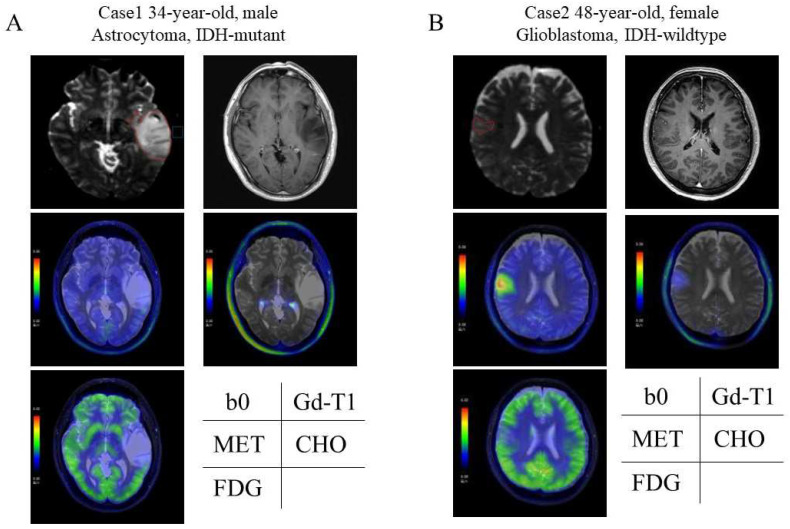
Representative Cases. Each ROI was drawn manually to surround the high-intensity area, with the maximum diameter in the b0 image. Case 1 (**A**): a 34-year-old male, astrocytoma, IDH mutant. The tumor invaded the insula. The FA value was 0.16, and the MD value was 1.53 × 10^−3^. The MET T/N ratio was 1.60, CHO T/N ratio was 1.26, and FDG T/N ratio was 0.76. The total points, including age, were 1, given by high MET accumulation. Case 2 (**B**): A 48-year-old female with glioblastoma, IDH-wildtype. The tumor is located in the right frontal cortex. The FA value was 0.19, and the MD value was 1.00 × 10^−3^. The MET T/N ratio was 4.78, CHO T/N ratio was 2.77, and FDG T/N ratio was 1.16. The total points, including age, were 6, given by old age, high FA, low MD, and high MET, CHO, and FDG accumulation. b0, b0 image; CHO, ^11^C-choline; FA, fractional anisotropy; FDG, ^18^F-fluorodeoxyglucose; Gd-T1, gadolinium-enhanced T1 weighted image; IDH, isocitrate dehydrogenase; MD, mean diffusivity; MET, ^11^C-methionine; T/N ratio, tumor/normal brain ratio.

**Table 1 cancers-16-01543-t001:** Summary of the non-contrast-enhanced cases.

	Astrocytomas, IDH-mut *n* = 55	Glioblastomas, IDH-wt *n* = 27	*p*-Value
Age (years), mean (SD)	38.0 (11.0)	52.1 (15.5)	<0.0001
Male, *n* (%)	32 (58.2%)	10 (37.0%)	n.s.
Intervals from MRI and PET examination to the operation (days), mean (SD)	47.9 (45.1)	71.6 (50.4)	0.008
Distribution, *n* (%)			
Single lobe	37 (67.3%)	16 (59.3%)	n.s.
Multiple lobes	17 (30.9%)	10 (37.0%)	n.s.
Insula	20 (36.4%)	11 (40.7%)	n.s.
Thalamus	1 (1.8%)	5 (18.5%)	0.006
Corpus callosum	11 (20.0%)	4 (14.8%)	n.s.
Reaching the contralateral hemisphere	1 (1.8%)	4 (14.8%)	0.018
Adjacent to the ventricle walls	22 (40.0%)	18 (66.7%)	0.022

IDH-mut, isocitrate dehydrogenase mutant; IDH-wt, isocitrate dehydrogenase wildtype; S.D., standard deviation; n.s., not significant.

## Data Availability

Data supporting the findings of this study will be available upon request.

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
