# Peer review of "Predicting Isocitrate Dehydrogenase Status in Non-Contrast-Enhanced Adult-Type Astrocytic Tumors Using Diffusion Tensor Imaging and 11C-Methionine, 11C-Choline, and 18F-Fluorodeoxyglucose PET"

_cancers, 2024, doi:10.3390/cancers16081543_

Round 1
Reviewer 1 Report
Comments and Suggestions for Authors
I commend the authors for a well-written paper that contributes to defining the role of nuclear medicine in the new diagnosis of gliomas. I believe the work is of interest to the journal and, after minor revisions, could be accepted for publication. Here are my concerns:
-
1) Title: While the definition of "astrocytic tumors" seems appropriate (adding "adult-type" for specificity), the phrase "using MRI and PET" is too simplistic and does not reflect the strengths of the work. If, as I understand, all patients underwent PET with three tracers, this is a significant strength of the article and should be highlighted. Similarly, regarding MRI, I would focus on the execution of DTI.
-
2) This leads to my second, and larger, concern: from the Materials and Methods, I understand that all patients underwent PET with three tracers. I am surprised that the ethics committee approved this, but if so, it is a major strength of the study. I also infer that PET acquisitions were made in the same session with all three tracers? And at what interval from the surgical intervention? I would specify this in the Materials and Methods. For the rest, the Materials and Methods seem clearly defined, and I believe that the subpopulation the study focuses on is indeed of greatest interest for the application of this multidiagnostic tool.
-
3) In the Discussion, I would provide more detailed differences between amino acid PET and FDG PET, citing a recent review published in this journal specifically addressing amino acid PET. (doi: 10.3390/cancers15010090)
-
-
4) In the Results, I would be very cautious in stating that thalamic location and reaching the contralateral hemisphere predict an IDH wild type because the absolute numbers on which statistical significance is based are very small. I would not cite them in the abstract as if they were a "strong" result, giving more space to diffusivity parameters.
Author Response
April 16, 2024
Prof. Dr. Samuel C. Mok
Editor-in-Chief
Cancers
Dear Dr. Samuel:
We greatly appreciate your suggestions and comments. We are returning herewith a manuscript revised according to the reviewer's comments. We hope that the revised manuscript is now acceptable for publication.
We would like to answer the reviewer's comment/suggestion point by point, as shown below. The following are answers to the suggestions.
Answers to Reviewer #1
- About the title
Thank you for your comments. We changed the more accurate title to clarify our study.
- About the timing of three tracers PET procedure and interval from MRI and PET examination to surgery
Thank you for your valuable questions.
We routinely perform three tracers PET for patients with brain tumors in one day because each tracer brings unique findings indicating biological behavior.
The intervals from the MRI and PET examination to the operation were 47.9 days in patients with astrocytomas and 71.6 days in those with glioblastomas. We added the information in the Result sections and Table 1.
Page 4, Line 162 – 163
The intervals from PET and MRI examination to the operation were shorter in the patients with astrocytomas than in the glioblastomas (p=0.008).
- About the differences between FDG-PET and amino acid PET
Thank you for your appropriate suggestion. We added the sentences about it.
Page 9, Line 332 - 336
Recently, revealing the effectiveness compared with FDG, other amino acid tracers using 18F, such as fluoroethyltyrosine or fluorodihydroxyphenylalanine, are expected to define the tumor extent or malignancy in glioma practice. L-amino acid transporters are broadly expressed in tumor tissue, which is not in normal brain tissues [28].
- About the description of MRI parameters in the Results section
Thank you for your essential suggestion about MRI parameters. The MRI findings indicate significant differences between astrocytomas and glioblastomas, identified by statistical methods from our dataset. Any inappropriate procedure did not qualify for this result. So, we thought it should be accepted as a fact, but we did not emphasize it more than necessary.
We appreciate your valuable suggestions, which will help readers understand our report. We are looking forward to hearing from you.
Sincerely,
Shoji Yasuda
630 Shimo-Kobi, Kobi-cho, Minokamo, 505-0034, Japan
Department of Neurosurgery and Chubu Medical Center for Prolonged Traumatic Brain Dysfunction, Chubu Neurorehabilitation Hospital,
Tel: 0574-24-2233
FAX: 0574-24-2230
E-mail: show_jis@yahoo.co.jp

Reviewer 2 Report
Comments and Suggestions for Authors
This manuscript describes clearly an advance in the diagnosis of IDH status in astrocytomas/glioblastomas using MRI and PET imaging in non-contrast enhanced gliomas.
In the abstract, the authors claim that their findings indicate that IDH-wt are more malignant than IDH-mut tumors. This is common knowledge and should be reworded in the abstract, i.e. these findings reflect or confirm previous findings.
Although limitations are mentioned, a statement could be provided to comment on the number of cases investigated which might be the biggest limitation. Also, the stringent exclusion criteria makes is difficult to translate these findings into the clinical practice.
Figure 3 and the scoring system is unclear and not well presented. Please improve the figure quality i.e. by using colours and present a table describing the scores in a comprehensive way.
Figure 4 the case descriptions should be set above the images.
Author Response
April 16, 2024
Prof. Dr. Samuel C. Mok
Editor-in-Chief
Cancers
Dear Dr. Samuel:
We greatly appreciate your suggestions and comments. We are returning herewith a manuscript revised according to the reviewer's comments. We hope that the revised manuscript is now acceptable for publication.
We would like to answer the reviewer's comment/suggestion point by point, as shown below. The following are answers to the suggestions.
Answers to Reviewer #2
1 About the simple summary
According to your valuable suggestions, we changed the expression "these findings might indicate higher malignancy…" to "these findings reflected higher malignancy…".
- About the limitation of this study
Thank you for your valuable opinion. The predicting tool for glioma generally faces problems caused by the rare population. Our study suggested a solution that understanding biological behavior connects with radiographical findings. Our method is not the only one that can predict IDH status, so we hope to develop a more accurate method to predict molecular diagnosis in glioma.
- About Figure 3 and 4
Thank you for your kind suggestion. We improved each figure.
We appreciate your valuable suggestions, which will help readers understand our report. We are looking forward to hearing from you.
Sincerely,
Shoji Yasuda
630 Shimo-Kobi, Kobi-cho, Minokamo, 505-0034, Japan
Department of Neurosurgery and Chubu Medical Center for Prolonged Traumatic Brain Dysfunction, Chubu Neurorehabilitation Hospital,
Tel: 0574-24-2233
FAX: 0574-24-2230
E-mail: show_jis@yahoo.co.jp
